

# Validation of *Plasmodium falciparum* deoxyhypusine synthase as an antimalarial target

Aiyada Aroonsri[1], Navaporn Posayapisit[1], Jindaporn Kongsee[2],
Onsiri Siripan[1,3], Danoo Vitsupakorn[1], Sugunya Utaida[2],
Chairat Uthaipibull[1], Sumalee Kamchonwongpaisan[1] and
Philip J. Shaw[1]

[1] Protein-Ligand Engineering and Molecular Biology Laboratory, Medical Molecular Biology
Research Unit, National Center for Genetic Engineering and Biotechnology (BIOTEC),
National Science and Technology Development Agency (NSTDA), Pathum Thani, Thailand
[2] Department of Biotechnology, Faculty of Science and Technology, Thammasat University,
Pathum Thani, Thailand
[3] Present address: Fisheries Industrial Technology Research and Development Division,
Department of Fisheries, Bangkok, Thailand

Corresponding author
Philip J. Shaw, philip@biotec.or.th

## ABSTRACT

**Background:** Hypusination is an essential post-translational modification in eukaryotes. The two enzymes required for this modification, namely deoxyhypusine synthase (DHS) and deoxyhypusine hydrolase are also conserved. *Plasmodium falciparum* human malaria parasites possess genes for both hypusination enzymes, which are hypothesized to be targets of antimalarial drugs.

**Methods:** Transgenic *P. falciparum* parasites with modification of the PF3D7_1412600 gene encoding *Pf*DHS enzyme were created by insertion of the *glmS* riboswitch or the M9 inactive variant. The *Pf*DHS protein was studied in transgenic parasites by confocal microscopy and Western immunoblotting. The biochemical function of *Pf*DHS enzyme in parasites was assessed by hypusination and nascent protein synthesis assays. Gene essentiality was assessed by competitive growth assays and chemogenomic profiling.

**Results:** Clonal transgenic parasites with integration of *glmS* riboswitch downstream of the *Pf*DHS gene were established. *Pf*DHS protein was present in the cytoplasm of transgenic parasites in asexual stages. The *Pf*DHS protein could be attenuated fivefold in transgenic parasites with an active riboswitch, whereas *Pf*DHS protein expression was unaffected in control transgenic parasites with insertion of the riboswitch-inactive sequence. Attenuation of *Pf*DHS expression for 72 h led to a significant reduction of hypusinated protein; however, global protein synthesis was unaffected. Parasites with attenuated *Pf*DHS expression showed a significant growth defect, although their decline was not as rapid as parasites with attenuated dihydrofolate reductase-thymidylate synthase (*Pf*DHFR-TS) expression. *Pf*DHS-attenuated parasites showed increased sensitivity to $N^1$-guanyl-1,7-diaminoheptane, a structural analog of spermidine, and a known inhibitor of DHS enzymes.

**Discussion:** Loss of *Pf*DHS function leads to reduced hypusination, which may be important for synthesis of some essential proteins. The growth defect in parasites with attenuated *Pf*DHS expression suggests that this gene is essential. However,

the slower decline of *Pf*DHS mutants compared with *Pf*DHFR-TS mutants in competitive growth assays suggests that *Pf*DHS is less vulnerable as an antimalarial target. Nevertheless, the data validate *Pf*DHS as an antimalarial target which can be inhibited by spermidine-like compounds.

## INTRODUCTION

The incidence of malaria has declined around the world in recent years, with a 21% reduction reported worldwide during the period 2010–2015 (*World Health Organization, 2016*). Programs to eliminate the disease in endemic countries could be thwarted by evolving *Plasmodium falciparum* parasite resistance to artemisinin-combination therapies that are widely used to treat and prevent infections (*Woodrow & White, 2017*). New drugs with novel modes of action are needed to combat drug-resistant parasites. Identification of novel drug targets should accelerate the development of such antimalarials.

Polyamines are nitrogenous base compounds that are essential for cellular proliferation and development. Malaria parasites synthesize large amounts of polyamines, in which spermidine is a major metabolite present in all stages of development (*Teng et al., 2009*). Moreover, several different polyamine analogues possess antimalarial activity, suggesting that polyamine metabolism constitutes novel drug targets (reviewed in *Clark et al. (2010)*). One of the main uses of spermidine in eukaryotes is for hypusination of translation initiation factor 5A (eIF5A) protein; two enzymes are required for this post-translational modification, namely deoxyhypusine synthase (DHS) and deoxyhypusine hydrolase (reviewed in *Park et al. (2010)*). *P. falciparum* possesses a single gene encoding eIF5A, and functional studies of the parasite eIF5A protein have shown that it is a substrate for hypusination (*Molitor et al., 2004*). Malaria parasites also possess canonical enzymes for hypusination of eIF5A, and the *P. falciparum* DHS enzyme (*Pf*DHS) uses eIF5A protein as a substrate for incorporation of spermidine (*Kaiser et al., 2007*). The enzymatic activity of *Pf*DHS is inhibited by $N^1$-guanyl-1,7-diaminoheptane (GC7), a known inhibitor of human DHS enzyme (*Kaiser et al., 2007*). Hypusination of eIF5A by *Pf*DHS is likely to be essential since *P. falciparum* is moderately sensitive to growth inhibition by GC7 (*Kaiser et al., 2001*) and no insertions of *piggyBac* transposon within the *Pf*DHS gene are tolerated (*Zhang et al., 2018*). The orthologous gene encoding DHS enzyme in the murine malaria parasite *P. berghei* is essential, since clonal transgenic *P. berghei* parasites with knockout of the DHS gene cannot be isolated (*Kersting et al., 2016*), and *P. berghei* DHS knockout parasites have a severe growth defect causing them to rapidly disappear from host animals co-infected with other transgenic parasites (*Bushell et al., 2017*).

In this study, we investigated *Pf*DHS function in transgenic *P. falciparum* parasites. The *glmS* riboswitch tool (*Prommana et al., 2013*) was used to attenuate *Pf*DHS

expression. Attenuation of *Pf*DHS expression led to defects in hypusination of eIF5A and growth of transgenic parasites. Moreover, attenuation of *Pf*DHS expression specifically sensitized parasites to GC7, a known inhibitor of *Pf*DHS enzyme activity.

# MATERIALS AND METHODS

## Ethics statement

Human erythrocytes were obtained from donors after providing informed written consent, following a protocol approved by the Ethics Committee, National Science and Technology Development Agency, Pathum Thani, Thailand, document no. 0021/2560.

## Construction of DNA transfection vectors

A total of 1,493 bp of homologous targeting sequence from the PF3D7_1412600 gene encoding deoxyhypusine synthase (*Pf*DHS) was PCR-amplified from *P. falciparum* strain 3D7 genomic DNA using primers pGFP_glmS_DHSF and GFP_glmS_DHSR (Table S1) and Phusion® DNA polymerase (Thermo Fisher Scientific, Waltham, MA, USA) following the manufacturer's recommendations. The pGFP_*glmS* and pGFP_M9 plasmids carrying *glmS* riboswitch element and the M9 inactive variant, respectively, (*Prommana et al., 2013*) were first modified to remove unnecessary hsp86 promoter and REP20 sequences by digestion with AflII and BglII. The digested plasmids were religated to create Sim_ pGFP_*glmS* and Sim_pGFP_M9 vectors. The *Pf*DHS targeting sequence was cloned into NheI and KpnI (New England Biolabs [NEB], Ipswich, MA, USA) digested vectors using a Gibson assembly kit (NEB). A total of 960 bp of homologous targeting sequence from the PF3D7_1364900 gene encoding ferrochelatase (*Pf*FC) was PCR-amplified from *P. falciparum* strain 3D7 genomic DNA using primers PfFCSacIIF and PfFCKpnIR (Table S1) and GoTaq® DNA polymerase (Promega Corporation, Madison, WI, USA) following the manufacturer's recommendations. The *Pf*FC targeting sequence DNA was digested with SacII and KpnI restriction enzymes (NEB) and cloned into pGFP_*glmS* plasmid (*Prommana et al., 2013*) digested with the same enzymes.

## *Plasmodium falciparum* culture and DNA transfection

*Plasmodium falciparum* strain 3D7 (reference strain) and transgenic parasite derivatives were cultured in vitro following the standard method (*Trager & Jensen, 1976*), with the modification of 0.5% Albumax I (Gibco™, Thermo Fisher Scientific, Waltham, MA, USA) replacing human serum (*Cranmer et al., 1997*). Human O+ erythrocytes were obtained from donors after obtaining their written informed consent. 2% hematocrit was used for most parasite cultures, with slightly higher hematocrit (up to 4%) used during blasticidin selection steps of DNA transfection. Parasites were synchronized to ring stages by sorbitol treatment (*Lambros & Vanderberg, 1979*). Approximately 50 µg of plasmid DNA was used for each parasite transfection experiment. Transfection was performed by the method of direct transfection of infected host cells (*Wu et al., 1995*) or invasion of DNA-loaded erythrocytes (*Deitsch, Driskill & Wellems, 2001*). Transgenic parasites were selected by treatment with 2 µg/mL blasticidinS-HCl (Gibco™), which was added to parasite culture 48 h post-transfection. Parasites were cultured under drug

selection until resistant parasites emerged (3 weeks), after which the drug was removed and culture continued for 2 weeks. The drug on-off cycle (2 weeks each) was repeated to enrich for integrant parasites in the population. Integrants were detected by PCR assay. Genomic DNA samples were obtained for genotypic analysis of transgenic parasites using a Genomic DNA Mini Kit (Blood/Cultured Cell) (Geneaid Biotech, New Taipei City, Taiwan). Integration at the *Pf*DHS locus was checked using primers DHS_intF and 3UTRpbDT_glmSR (Table S1). PCRs contained 20 ng of genomic DNA, 500 nM of each primer, one unit of Phusion polymerase (Thermo Scientific™, Thermo Fisher Scientific, Waltham, MA, USA) and 2.5 mM MgCl$_2$. The reaction conditions were: 98 °C for 3 min followed by 35 cycles of 98 °C 10 s, 52 °C 30 s, and 72 °C for 90 s, and final extension at 72 °C for 5 min. The presence of transfected DNA in transfection experiments to modify the *Pf*FC gene was checked using primers BglIIPfFCF and 3UTRpbDTglmSR (Table S1). A control PCR to amplify the *Pf*FC coding region and thus verify template DNA quality was performed using primers BglIIPfFCF and PfFCKpnIR (Table S1). PCR assays of the *Pf*FC locus contained 20 ng of genomic DNA, 100 nM of each primer, one unit of GoTaq polymerase (Promega Corporation), and 2.5 mM MgCl$_2$. The reaction conditions were: 95 °C for 2 min followed by 30 cycles of 95 °C 45 s, 53 °C 45 s, and 62 °C for 2 min 30 s, and final extension at 62 °C for 5 min.

Clonal lines of integrant transgenic parasites were established by limiting dilution in 96-well culture plates. A single clonal line from each transfection experiment, verified as integrant, was randomly selected for further study. Clonal lines *Pf*DHS_*glmS* (active riboswitch) and *Pf*DHS_M9 (inactive riboswitch) with integration at the *Pf*DHS locus were obtained (Fig. S1). The clonal line *Pf*FC_*glmS* (active riboswitch) with integration at the *Pf*FC locus was obtained. Plasmid integration in clonal transgenic lines was verified by Southern blot (Figs. S1 and S2). A total of 20 μg samples of genomic DNA were digested overnight with restriction enzymes (NEB). BamHI and AflII enzymes were used for the *Pf*DHS locus and SpeI and HindIII enzymes were used for the *Pf*FC locus. Digested DNA samples were separated by electrophoresis and transferred by capillary action to Hybond N+ nylon membrane (GE Healthcare, Chicago, IL, USA). Probes were synthesized by PCR using primers pGFP_glmS_DHSF and GFP_glmS_DHSR for the *Pf*DHS locus, and PfFCSacIIF and PfFCKpnIR for the *Pf*FC locus (Table S1) as described above. Probe labeling, hybridization, and detection were performed using a DIG High Prime DNA labeling and Detection Starter Kit II (Roche Diagnostics, Basel, Switzerland) following the manufacturer's instructions. The integrant transgenic parasite line described in (Prommana et al., 2013), referred to here as *Pf*DHFR-TS_*glmS*, was used for growth studies. This parasite has a modified PF3D7_0417200 gene encoding *P. falciparum* dihydrofolate reductase-thymidylate synthase (*Pf*DHFR-TS) with integration of GFP and *glmS* riboswitch sequences.

## Confocal microscopy

Specimens of transgenic parasites *Pf*DHS_*glmS*, *Pf*FC_*glmS*, *Pf*DHFR-TS_*glmS*, and 3D7 parental parasites were analyzed on a model FV 1,000D IX81 confocal laser scanning microscope (Olympus, Shinjuku, Japan). A 100× oil immersion objective lens (1.4 NA)

was used. Parasite mitochondria were stained with 1 mM Mitotracker (Invitrogen™, Carlsbad, CA, USA; Thermo Fisher Scientific, Waltham, MA, USA) for 45–60 min at 37 °C. Parasite nuclei were stained with Hoechst 33342 (Invitrogen™) diluted 1:1,000 in RPMI medium (Gibco™) for 5 min at 37 °C. GFP signal was detected with an Argon laser 488 nm (500 nm excitation/600 nm emission; laser power 15%; high detector sensitivity 741 V; gain = 1 and offset = 12%), Mitotracker signal was detected with a yellow diode laser 559 nm (575 nm excitation/675 nm emission; laser power 15%; high detector sensitivity 641 V; gain = 1 and offset = 0%, and Hoechst signal was detected with a UV laser diode 405 nm (425 nm excitation/475 nm emission; laser power 10%; high detector sensitivity 615 V; gain = 1, and offset = 14%). Images were obtained using a scan speed of 10.0 μs/pixel and were analyzed using FV10-ASW 3.0 Viewer software (Olympus, Shinjuku, Japan).

## Western immunoblot of *Pf*DHS-GFP protein

A total of 15 mL cultures of ring-stage *Pf*DHS_*glmS* and *Pf*DHS_M9 synchronized parasites at 5% parasitemia and 3% haematocrit were treated with 0, 1.25, 2.5, and 5.0 mM GlcN for 24 h and harvested. Parasites were liberated from host cells by saponin lysis and washed with 1× PBS buffer (Thermo Scientific™) containing 1× EDTA-free protease inhibitor cocktail (Sigma-Aldrich, Merck KGaA, Germany) and 0.7 μg/mL pepstatin (Sigma). Parasite proteins were extracted by freeze-thaw rupture. Protein lysate was diluted in NuPAGE™ LDS sample buffer (4×, Thermo Scientific™) and stored at −80 °C. A 10 μg sample from each protein lysate was separated in NuPAGE 4–12% Bis-Tris Protein Gel (Invitrogen™) with 1× NuPAGE MOPS SDS running buffer (Invitrogen™). Proteins were transferred for 1.5 h at 30 V onto Immobilon-P PVDF membrane (Merck-Millipore, Merck KGaA, Germany) in 1× NuPAGE transfer buffer (Invitrogen™) by using a XCell II Blot system (Invitrogen™). Total protein was detected by Ponceau-S (Sigma) staining and an image of the stained membrane was captured on a flatbed scanner. After scanning, the Ponceau-S stain was removed from the membrane by washing with water. The destained membrane was incubated in blocking solution (5% non-fat skimmed milk in 1× TBST buffer; 10 mM Tris-base, 15 mM NaCl, pH 8.0, 0.05% Tween 20) for 1 h. *Pf*DHS-GFP protein was immunodetected with 1:5,000 dilutions of rabbit anti-GFP polyclonal antibody (#PA1-19431; Thermo Scientific™) and goat anti-rabbit IgG antibody conjugated with HRP (#SC-2004; Santa-Cruz Biotechnology, Dallas, TX, USA). Proteins were detected by chemiluminescence using Pierce ECL Western Blotting Substrate (Thermo Scientific™). The intensity of the *Pf*DHS-GFP band was measured by densitometry from the scanned image of the exposed *X*-ray film using the Image J program (*Schneider, Rasband & Eliceiri, 2012*). The immunodetected protein band signals were normalized to total protein signal in each lane. The percent relative intensities of *Pf*DHS-GFP are intensities in GlcN treatment conditions relative to the untreated control.

## Western immunoblot of *Pf*DHFRTS-GFP and *Pf*FC-GFP protein

*Pf*FC_*glmS*, *Pf*DHFR-TS_*glmS*, and 3D7 parasites were cultured and synchronized as described above and harvested at the trophozoite stage. Parasites were liberated from erythrocytes by saponin lysis. Parasites were resuspended in 1× NuPAGE LDS sample

buffer in a ratio of parasite cell/buffer volume of $2 \times 10^6$ parasites/mL and incubated at 95 °C for 10–15 min for protein extraction. Insoluble material was separated by centrifugation. Protein samples from $1 \times 10^6$, $5 \times 10^6$, $10 \times 10^6$, or $25 \times 10^6$ parasites were separated by electrophoresis as described above for *Pf*DHS-GFP Western immunoblot. Precision Plus Protein™ Dual Color Standards (Bio-Rad, Hercules, CA, USA) were used as a protein ladder. Proteins were transferred to Immobilon-FL PVDF membrane (Merck) as described above for *Pf*DHS-GFP Western immunoblot. After transfer, membranes were stained with REVERT™ Total Protein Stain (LI-COR Biosciences, Lincoln, NE, USA). Total protein was visualized using an Odyssey® CLx system (LI-COR) in the 700 nm channel. After imaging, REVERT stain was removed by washing with REVERT Reversal Solution (LI-COR) and water. The membrane was then incubated in Odyssey blocking buffer (TBS) (LI-COR) for 1 h with 80 rpm shaking. The blocked membrane was incubated in Odyssey blocking buffer (TBS) with 0.2% (v/v) Tween 20 and 1:2,000 diluted rabbit anti-GFP polyclonal antibody (#G1544; Sigma) overnight with 80 rpm shaking. The membrane was then washed with $1\times$ TBST ($1\times$ TBS with 0.05% Tween 20) and incubated in Odyssey blocking buffer (TBS) with 0.2% Tween 20 and IRDye 800CW 1:20,000 diluted goat anti-rabbit IgG (LI-COR) for 1 h at room temperature in the dark. The membrane was scanned in the 800 nm channel and images were analyzed using Image Studio Software (LI-COR).

## Hypusination assay

A total of 10 mL *P. falciparum* sorbitol-synchronized cultures at approximately 2% hematocrit and 1% ring-stage parasitemia were treated with 0, 2.5, or 5.0 mM GlcN for 72 h. Parasites were harvested and liberated from erythrocytes by saponin lysis. Parasite pellets were resuspended in 1% Triton-X100 in $1\times$ PBS buffer and incubated at 4 °C for 20 min to extract protein. Protein samples from approximately $5 \times 10^6$ parasites were separated by electrophoresis as described above for *Pf*DHS-GFP Western immunoblot. Precision Plus Protein Kaleidoscope Prestained Protein Standards (Biorad) were used as a protein ladder. Proteins were transferred to Immobilon-FL PVDF membrane (Merck) and processed before detection with Odyssey CLx as described above. The blocked membrane was incubated in Odyssey blocking buffer (TBS) with 0.2% (v/v) Tween 20 and 1:5,000 diluted rabbit anti-hypusine polyclonal antibody (#ABS1064; Merck) for 1 h with 80 rpm shaking. The membrane was then washed with $1\times$ TBST ($1\times$ TBS with 0.05% Tween 20) and incubated in Odyssey blocking buffer (TBS) with 0.2% Tween 20 and IRDye 680RD Goat anti-Rabbit IgG (LI-COR) for 1 h at room temperature in the dark. The membrane was scanned in the 700 nm channel and images were analyzed using Image Studio Software (LI-COR). Total protein loading in each lane was quantified by the sum of pixels minus background in rectangular objects spanning polypeptides 15–150 kDa. The major band of hypusinated protein signal assumed to be *Pf*eIF5A (17.6 kDa) was normalized to the total protein signal in each lane and expressed relative to the control lane (zero mM GlcN treatment).

## Nascent protein synthesis (puromycilation) assay

Nascent protein synthesis was assessed using the puromycilation assay (*Schmidt et al., 2009*). This assay is based on the incorporation of the translation inhibitor puromycin into

nascent peptide chains by actively translating ribosomes (*Nathans, 1964*). A previous report showed that this assay is suitable for *P. falciparum* (*McLean & Jacobs-Lorena, 2017*). Synchronized *P. falciparum* cultures were treated with GlcN for 72 h as described above for hypusination assay. After GlcN treatment, puromycin (Sigma) was added to a final concentration of 5 μM and parasites were incubated at 37 °C for an additional 10 min. The parasitized red blood cells were then washed with incomplete medium and parasites were liberated from erythrocytes by saponin lysis. Protein samples were obtained from approximately $5 \times 10^6$ parasites, separated by electrophoresis, transferred to PVDF membrane and processed before detection as described above for hypusination assay. After blocking, the membrane was incubated in Odyssey blocking buffer (TBS) with 0.2% (v/v) Tween 20 and 1:10,000 diluted mouse anti-puromycin monoclonal antibody clone 12D10 (Merck) for 1 h with 80 rpm shaking. The membrane was washed with 1× TBST (1× TBS with 0.05% Tween 20) and incubated in Odyssey blocking buffer (TBS) with 0.2% Tween 20, IRDye 800CW Goat anti-Mouse IgG (LI-COR) for 1 h at room temperature in the dark. The membrane was washed with 1× TBST (1× TBS with 0.05% Tween 20) and 1× TBS buffer before scanning in the 800 nm channel. Total protein and puromycilated peptides in each lane were quantified by the sum of pixels minus background in rectangular objects spanning polypeptides 15–150 kDa. The puromycilated peptide signal was normalized to the total protein signal in each lane and expressed relative to the control lane (zero mM GlcN treatment). The puromycilation assay was validated using *P. falciparum* synchronized cultures at mostly trophozoite stage. Parasites were pre-treated with growth-inhibitory compounds cycloheximide (1, 10, or 100 μM); dihydroartemisinin (0.01, 0.10, or 1 μM), and GC7 (50, 100, or 1,000 μM) for 1 h prior to puromycin exposure.

## Competitive growth assay

*P. falciparum* transgenic parasite line *Pf*FC_*glmS* was used as a control. The growth of a test transgenic line with riboswitch element integrated at a putative essential gene was normalized to that of the *Pf*FC_*glmS* control. The control and test transgenic lines were first cultured separately and synchronized as described above. Ring-stage synchronized parasites were diluted to approximately 0.5% parasitemia. A new culture was established by combining control and test transgenic parasite cultures (5 mL of each) into the same culture plate. Samples were taken from the parasite co-culture every 4 days for 21 days (10 growth cycles). At each sampled time-point, the culture reached a parasitemia of approximately 2.5%, consisting of mostly trophozoite stage parasites. The culture was then diluted to approximately 0.1% parasitemia in fresh medium. The parasite culture plates were placed in a refrigerator on "ring" days for 5–7 h to maintain high synchronization (*Yuan et al., 2014*). Parasite pellets were harvested and used for genomic DNA extraction. The co-culture was conducted under standard and gene attenuation (2.5 or 5.0 mM GlcN inducer added) conditions, in which fresh GlcN was added after each time-point. The effect of GlcN on development of control parasites 3D7 and *Pf*FC_*glmS* was assessed by counting ring and trophozoite stage parasites separately from Giemsa-stained thin smears.

## Quantitative PCR assay of transgenic parasite ratios in competitive growth experiments

Primers for quantitative PCR (qPCR) assays were designed to amplify specific discriminatory sequences in transgenic parasites tested in competitive growth assays. The discriminatory sequences spanned the fusion boundary between the 3′ end of the modified target gene and the GFP coding region. The LDH-F and LDH-R primers (Table S1; amplicon 221 bp) were also designed to amplify the PF3D7_1324900 (L-lactate dehydrogenase) gene, which is present at single copy in all parasites and is used for normalization of template DNA input. qPCR experiments were performed using a CFX96 Touch Real-Time PCR Detection System (Biorad) and SsoFast EvaGreen Supermix (Biorad) in 20 μL reaction volumes, as recommended by the manufacturer. All primer pairs performed with similar efficiency (96–103%, linear regression $R^2 > 0.99$). Amplicons of the expected size were obtained only from the expected template genomic DNA, as assessed by agarose gel electrophoresis and melt-curve analysis. The *Pf*DHS_*glmS* and *Pf*DHS_M9 parasites were quantified from the DHS-GFP amplicon (188 bp, primers DHS-F and DHS-R; Table S1), the *Pf*DHFR-TS-*glmS* parasite (*Prommana et al., 2013*) was quantified from the TS-GFP amplicon (190 bp, primers DHFRTS-F and DHFRTS-R; Table S1), and the *Pf*FC-*glmS* parasite was quantified by the FC-GFP amplicon (125 bp, primers FC-F and FC-R; Table S1). The quantitative range of qPCR assays was determined using purified genomic DNA extracted from transgenic parasites that were mixed in different ratios (Fig. S3).

## Dose-response growth inhibition assays

The growth of parasites was monitored under different concentrations of growth inhibitors in dose-response assays as described previously (*Aroonsri et al., 2016*). To assess the effect of GlcN on growth inhibition, parasites were co-treated with 2.5 mM GlcN. The growth inhibitory compounds tested included $N^1$-guanyl-1,7-diaminoheptane (GC7; Sigma), cycloheximide (CYC; Sigma), and pyrimethamine (PYR; Sigma). Stock solutions of growth inhibitors were prepared fresh for each experiment, in which compounds were dissolved in DMSO (CYC and PYR) or culture medium (GC7). Compounds were diluted in culture medium, in which the maximum concentration of DMSO did not exceed 0.1%. Control wells with no growth inhibitory compound contained 0.1% DMSO.

## Statistical analysis

All statistical analysis was performed using R 3.4.3 (*R Core Team, 2017*). For analysis of Western immunoblot, hypusination, and puromycilation assay data, Welch's two-tailed one-sample *t*-tests (*Welch, 1947*) were performed by testing the null hypothesis that sample means of test signal normalized to untreated control was not different from one. Tests with $P < 0.05$ were considered significant.

The lme4 package (*Bates et al., 2015*) was used to perform a linear mixed effects analysis of the relationship between parasite growth and time in competitive growth assays. For validation experiments with 3D7 parental and *Pf*FC-*glmS* control transgenic lines, the percentage of ring or trophozoite-infected cells was taken as the growth variable.

In experiments with *Pf*DHFR-TS-*glmS*, *Pf*DHS_*glmS*, and *Pf*DHS_M9 transgenic parasites, growth relative to the *Pf*FC-*glmS* control transgenic parasite determined by qPCR was taken as the growth variable. GlcN treatment (with doses as factors) was modeled as a fixed effect. Individual parasite cultures grown on different days were modeled as random effects, with random intercepts included in the model for the effect on growth over time. Linear models were constructed from the data using maximum likelihood. The null model was that growth varies as a function of time. The full model was that growth varies as a function of time, with GlcN treatment and its interaction with time as fixed effects. Significant differences in model fitting were assessed by likelihood ratio test, with $P < 0.05$ considered significant. Fitted models were plotted using the visreg package (*Breheny & Burchett, 2017*).

Normalized growth values from dose-response assays were analysed using the drc package (*Ritz & Streibig, 2005*). To allow proper comparison of $EC_{50}$ values between the −GlcN and +GlcN conditions, the maximum and minimum growth values were assigned as shared and constant between the two conditions, such that only two parameters (slope and $EC_{50}$) varied between the −GlcN and +GlcN conditions. $EC_{50}$ ratios, associated S.Es and *t*-statistics were calculated from the two-variable parameter fitted models using the EDcomp function in the drc package. $EC_{50}$ ratios were considered significant at $P < 0.001$.

## In silico modeling of *Pf*DHS protein structure and GC7 binding

A three-dimensional structure of the *Pf*DHS tetramer was constructed by the SWISS-MODEL Server (*Waterhouse et al., 2018*). The program selected the *X*-ray crystal structure of Form I human DHS complexed with NAD as the homology template (*Umland et al., 2004*); PDB ID: 1RLZ. Ligand binding site identification and characterization was performed using SiteMap version 3.6, Schrödinger, LLC, New York (*Halgren, 2009*). The grid type was set as coarse and other settings were default. The putative ball-and-chain motif encompassing residues Ile30–Pro45 of each *Pf*DHS subunit was removed prior to ligand binding site analysis. Ligand binding sites with scores greater than 1.0 were considered significant. Molecular docking of GC7 at the substrate-binding pocket was performed using Glide (*Friesner et al., 2004*). A grid receptor was generated around the binding pocket with addition of void volume around NAD cofactor. The structure of Form II human DHS complexed with NAD and GC7 (*Umland et al., 2004*); PDB ID: 1RQD was used for comparison of *Pf*DHS and human DHS substrate binding pockets.

## RESULTS

Clonal transgenic *P. falciparum* parasite lines *Pf*DHS_*glmS* and *Pf*DHS_M9 were established with integration of *glmS* riboswitch and the M9 inactive variant, respectively, at the *Pf*DHS encoding gene PF3D7_1412600. The DNA vectors used for integration also contain GFP gene sequence for C-terminal tagging of the target protein. Fluorescent parasites with GFP signal in the parasite cytoplasm were observed in ring, trophozoite, and schizont stages for the *Pf*DHS_*glmS* parasite (Fig. 1). Western immunoblotting with anti-GFP antibody revealed a species that migrated slightly larger than predicted for *Pf*DHS-GFP fusion protein (85.1 kDa) from *Pf*DHS_*glmS* and *Pf*DHS_M9 parasites

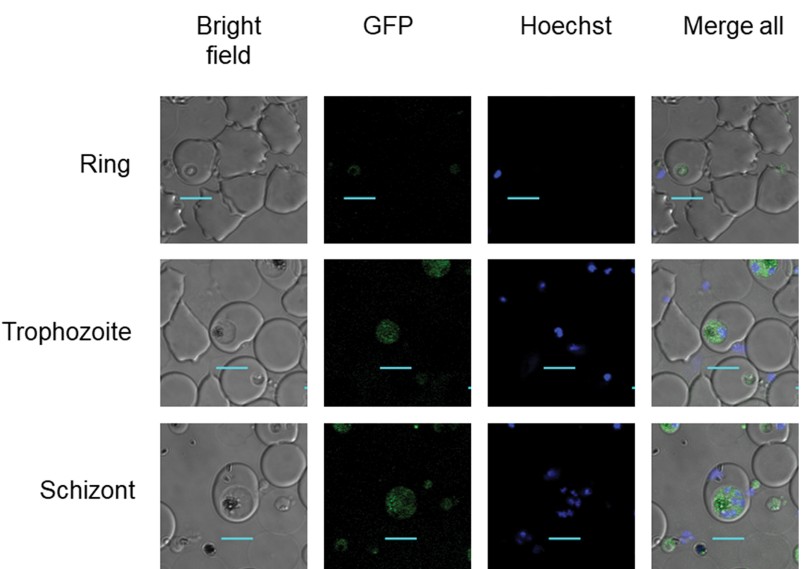

|  | Bright field | GFP | Hoechst | Merge all |
|---|---|---|---|---|

Ring

Trophozoite

Schizont

**Figure 1 *Pf*DHS protein localization in transgenic parasites.** Representative confocal microscopic images of *Pf*DHS_*glmS* parasites expressing GFP-tagged *Pf*DHS protein at ring, trophozoite, and schizont stages. Parasite nuclei were stained with Hoechst 33342. Composite images from merging Hoechst and GFP fluorescence signals with the Bright-field image are shown in the panels on the far-right. Scale bars = 5 μm.

(Fig. 2). *Pf*DHS-GFP protein was significantly attenuated in the *Pf*DHS_*glmS* parasite treated for 24 h with GlcN, with up to fivefold reduction observed with 5.0 mM GlcN. In contrast, no significant reduction of *Pf*DHS-GFP protein was observed with GlcN treatment in the *Pf*DHS_M9 parasite with an inactive riboswitch.

Since the primary function of DHS enzyme in eukaryotes is hypusination of eIF5A protein, hypusination was quantified using a commercial anti-hypusine antibody in transgenic parasites. A major protein species (<20 kDa) was detected by Western immunoblotting with this antibody which matches the predicted molecular weight of *P. falciparum* eIF5A (17.6 kDa) and is the approximately the same size as *Pfe*IF5A detected with anti-eIF5A antibodies (*Foth et al., 2008*). However, additional evidence, for example, peptide mapping is needed to confirm that this species is *Pfe*IF5A. GlcN treatment led to significantly reduced hypusinated protein signal in *Pf*DHS_*glmS* parasites treated with GlcN compared with untreated controls, although this effect was rather small with <30% mean reduction at 5.0 mM GlcN (Fig. 3). In contrast, GlcN treatment had no significant effect on hypusination in *Pf*DHS_M9 parasites. We infer from this result that reduction of *Pf*DHS expression leads to concomitant reduction of hypusinated protein. Hypusination is thought to be essential for the translation elongation function of this protein (*Park et al., 2010*). We tested whether reduced hypusination could affect protein synthesis by nascent protein synthesis (puromycilation) assay in *Pf*DHS_*glmS* and *Pf*DHS_M9 parasites. Nascent protein was quantified by the incorporation of puromycin into elongating peptide chains, detected as the signal from Western immunoblotting with anti-puromycin antibody. Short pre-treatments of parasites with lethal (>EC$_{50}$) concentrations of CYC and DHA known to cause arrest of protein synthesis in

**A**

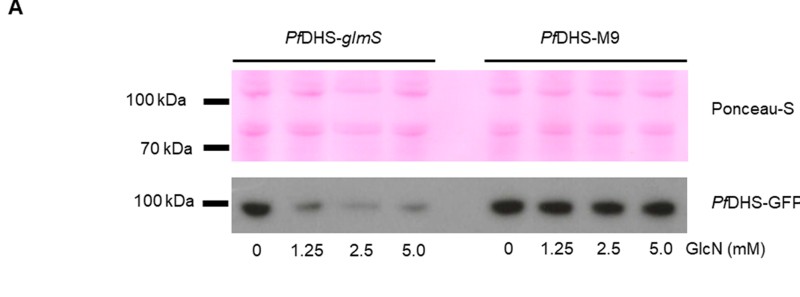

**B**

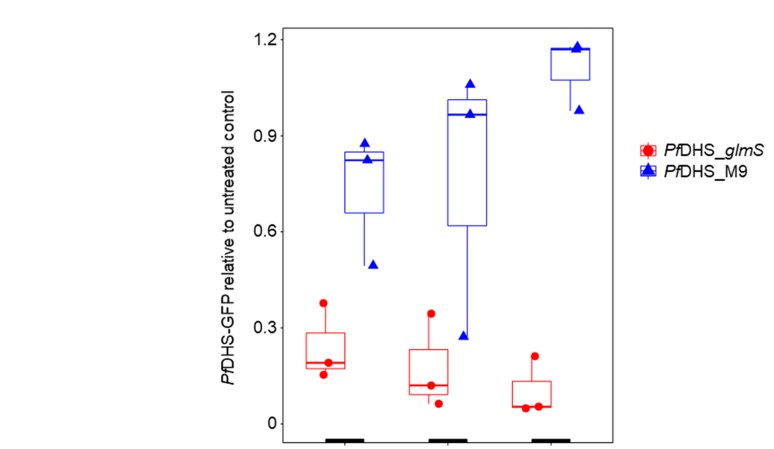

**Figure 2 Attenuation of *Pf*DHS expression in transgenic parasites.** (A) Representative Western immunoblot results from detection of GFP-tagged *Pf*DHS protein (*Pf*DHS-GFP) in *Pf*DHS_*glmS* and *Pf*DHS_M9 transgenic parasites. Soluble protein extracts were obtained from approximately $10 \times 10^6$ parasites treated for 24 h with 0, 1.25, 2.50, and 5.00 mM glucosamine (GlcN). Proteins were separated in 4–12% NuPAGE gel and transferred to PVDF membrane. Total protein was stained on the membrane using Ponceau-S (top panel) and *Pf*DHS-GFP protein was detected using anti-GFP antibody (bottom panel). The migrations of PageRuler Plus Prestained Protein ladder (Thermo Scientific, Waltham, MA, USA) standards are indicated on the left. The images are cropped for clarity. Full-length, uncropped blot images are shown in Fig. S4. (B) Quantified Western immunoblot results. The signal intensity of the *Pf*DHS-GFP protein band was normalized to the total protein and the *Pf*DHS-GFP protein signal intensity in the sample lane from untreated parasites. The data from three independent experiments are shown for each parasite line. *P*-values from one-sample *t*-tests: 0.0082 (*Pf*DHS_*glmS*, 1.25 mM GlcN); 0.011 (*Pf*DHS_*glmS*, 2.5 mM GlcN); 0.0035 (*Pf*DHS_*glmS*, 5.0 mM GlcN); 0.15 (*Pf*DHS_M9, 1.25 mM GlcN); 0.45 (*Pf*DHS_M9, 2.5 mM GlcN), and 0.24 (*Pf*DHS_M9, 5.0 mM GlcN).

*P. falciparum* (*Hoepfner et al., 2012*; *Rottmann et al., 2010*; *Shaw et al., 2015*; *McLean & Jacobs-Lorena, 2017*) led to markedly reduced puromycilation signal in both transgenic parasites, validating the assay (Fig. S5). Puromycilation signal was not significantly different in either parasite treated with GlcN for 72 h prior to puromycin labeling compared with parasites with no GlcN pre-treatment, suggesting that attenuation of *Pf*DHS expression has little effect on the global translation level during this period (Fig. 3). Moreover, GlcN-treated parasites do not show any gross morphological defect (Fig. S6) and proliferation over 72 h is unaffected by GlcN (Fig. S7). These data suggest that the slight reduction of hypusination in parasites with attenuated *Pf*DHS function is tolerated for short periods.
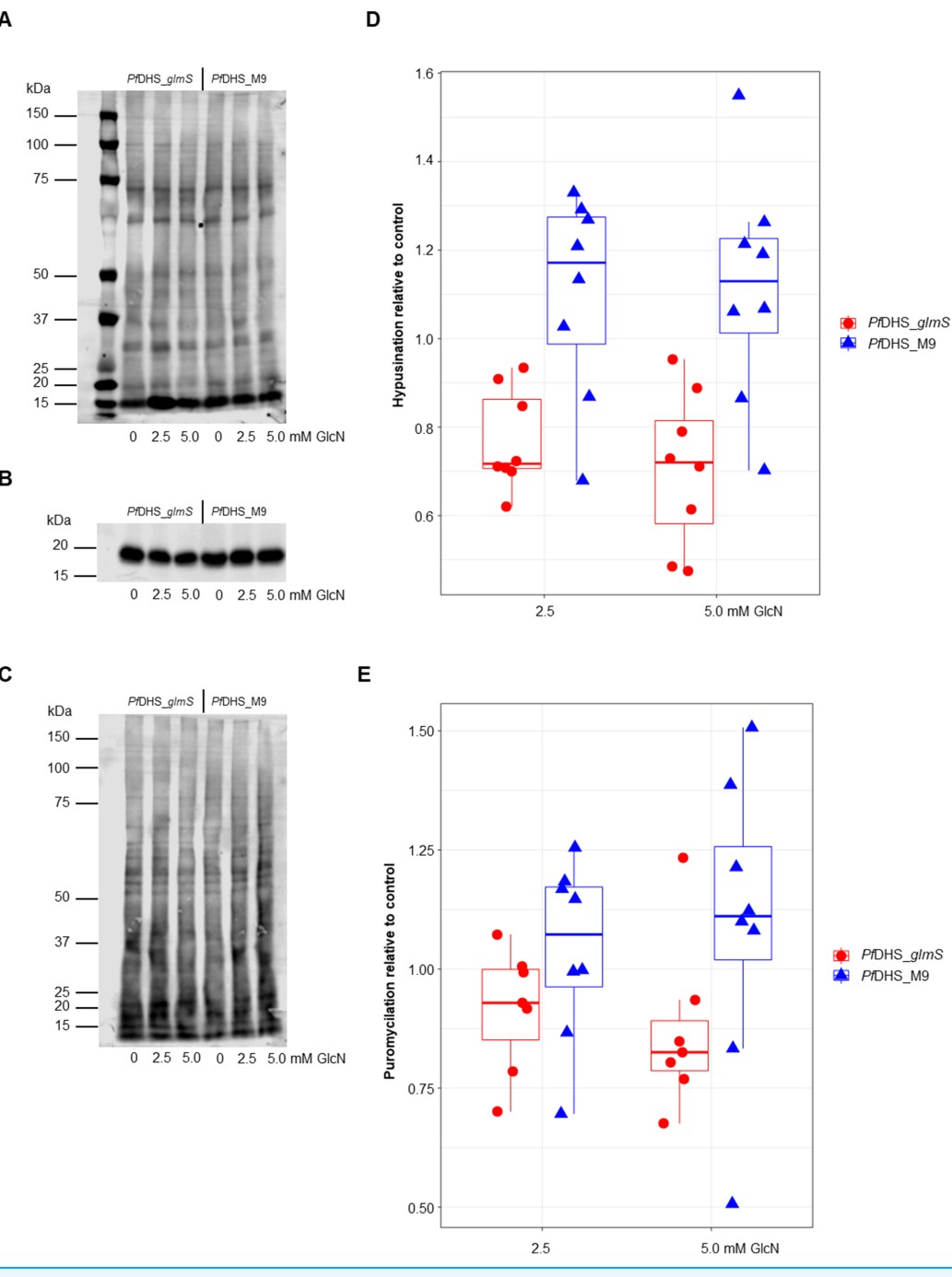

**Figure 3 Hypusination and nascent protein synthesis assays.** Ring-synchronized *Pf*DHS_*glmS* and *Pf*DHS_M9 transgenic parasites were treated with 0, 2.5, or 5.0 mM glucosamine (GlcN) for 72 h. For assays of nascent protein synthesis, parasites were subsequently treated with 5 μM puromycin for 10 min prior to harvesting of parasites. Protein was extracted from parasites and separated in 4–12% NuPAGE gel. Representative data are shown in parts (A–C). The images in parts (A) and (B) are cropped for clarity. Full-length, uncropped blot images are shown in Fig. S5. (A) Total protein stained with REVERT. (B) Hypusination assay results. A band corresponding to *Plasmodium falciparum* eIF5A (17.6 kDa) was detected with anti-hypusine polyclonal antibody. (C) Nascent protein synthesis assay. Puromycilated, nascent peptides were detected with anti-puromycin monoclonal antibody. (D) Quantified hypusination assay results from eight independent experiments. *P*-values from one-sample *t*-tests: 7.0e-4 (*Pf*DHS_*glmS*, 2.5 mM GlcN); 0.0020 (*Pf*DHS_*glmS*, 5.0 mM GlcN); 0.25 (*Pf*DHS_M9, 2.5 mM GlcN),

**Figure 3** (continued)

and 0.25 (*Pf*DHS_M9, 5.0 mM GlcN). (E) Quantified puromycilation assay results from seven (*Pf*DHS_*glmS*) and eight (*Pf*DHS_M9) independent experiments. *P*-values from one-sample *t*-tests: 0.13 (*Pf*DHS_*glmS*, 2.5 mM GlcN); 0.10 (*Pf*DHS_*glmS*, 5.0 mM GlcN); 0.58 (*Pf*DHS_M9, 2.5 mM GlcN), and 0.42 (*Pf*DHS_M9, 5.0 mM GlcN).               

Although attenuation of *Pf*DHS gene expression is not deleterious in the short term, prolonged loss of *Pf*DHS activity could lead to a growth defect. In the standard growth assay, parasite cultures are initiated at low parasite density, for example, 0.1% parasitemia, and growth assessed over the first two cycles (up to 96 h). For some essential genes, *glmS* riboswitch-mediated attenuation of expression can cause a significant growth defect under these conditions, for example, *Pf*DHFR-TS (*Prommana et al., 2013*), *Pf*PTEX150 (*Elsworth et al., 2014*), *Pf*RhopH2 (*Counihan et al., 2017*; *Ito, Schureck & Desai, 2017*) and *Pf*RhopH3 (*Ito, Schureck & Desai, 2017*). However, *glmS* riboswitch-mediated attenuation of other essential genes such as *Pf*PMV (*Sleebs et al., 2014*) and *Pf*FP3 (*Xie et al., 2016*) failed to show a growth defect in the standard growth assay. The inability to detect growth defects in the standard growth assay is due to factors such as insufficient attenuation, functional overlap/redundancy with related proteins and stage-specific target protein function. For growth assays longer than two cycles, dilution of cells is necessary to prevent overgrowth of controls. Transgenic parasites obtained by single cross-over recombination must be maintained under a drug selective regimen, for example, blasticidin, otherwise they can be outgrown by wild-type revertants. To account for possible growth confounding effect of the transgenic selective regimen, a control transgenic parasite (*Pf*FC_*glmS*) was created with integration of the *glmS* riboswitch at the ferrochelatase (*Pf*FC) gene. The *Pf*FC gene is dispensable during intra-erythrocytic stages, since the growth of clonal transgenic parasites with knockout of this gene is not significantly different from wild-type (*Ke et al., 2014*; *Sigala et al., 2015*). We could not assess localization or riboswitch-mediated attenuation of *Pf*FC-GFP protein in the *Pf*FC_*glmS* parasite, since no green fluorescent *Pf*FC_*glmS* parasites were observed by microscopy (Fig. S8), and the weak signal of putative *Pf*FC-GFP protein was difficult to distinguish from background in Western immunoblotting experiments (Fig. S9). We validated the *Pf*FC_*glmS* parasite as a control in growth experiments by assessing its growth in comparison with 3D7 wild-type parasite under different GlcN treatments. Over the course of 10 growth cycles, treatment with 2.5 or 5.0 mM GlcN has no significant effect on ring or trophozoite development for both 3D7 and *Pf*FC_*glmS* parasites (Fig. S10).

We established a competitive growth assay in which transgenic parasites are cultured for 10 growth cycles. The extended growth period in this assay allows for detection of latent defects that are apparent only after several growth cycles have elapsed. In this assay, a test transgenic parasite is co-cultured with the control *Pf*FC_*glmS* transgenic parasite. The growth of test transgenic parasite was assessed by measuring the test:control transgenic parasite ratio by qPCR. The use of the control *Pf*FC_*glmS* parasite allowed us to monitor growth without correction of dilution factors. Relative growth of the *Pf*DHS_*glmS*

parasite was significantly reduced under GlcN treatment (Fig. 4A). In contrast, GlcN had no significant effect on the relative growth of the *Pf*DHS_M9 parasite (Fig 4B). Competitive growth assay of the *Pf*DHFR-TS_*glmS* parasite revealed a more severe growth defect when treated with GlcN in which *Pf*DHFR-TS_*glmS* parasites declined more rapidly (Fig. 4C). The results from competitive growth assays showed that attenuation of *Pf*DHS expression causes a growth defect, indicating that this gene is likely to be essential. As a further test of this gene's essentiality and validation as an antimalarial target, chemogenomic profiling using antimalarial compounds was performed. In this approach, *glmS* riboswitch-mediated attenuation of target gene expression sensitizes the parasite to antimalarials acting on that target (*Aroonsri et al., 2016*). Chemogenomic profiling of transgenic parasites was performed with growth-inhibitory compounds differing in their mode of action (Fig. 4D). The *Pf*DHS_*glmS* parasite is significantly more sensitive to GC7, a known inhibitor of *Pf*DHS enzyme activity in vitro (*Kaiser et al., 2007*), when co-treated with GlcN. We noted though that the $\log_2$ $EC_{50}$ (–GlcN/+GlcN) ratio is less than one for the *Pf*DHS_*glmS* parasite. In contrast, the *Pf*DHS_M9 parasite is not significantly more sensitive to GC7 when co-treated with GlcN. GlcN co-treatment does not significantly increase the sensitivity of either parasite to control drugs which do not target the *Pf*DHS, including CYC, a known inhibitor of the ribosome (*Obrig et al., 1971*; *Schneider-Poetsch et al., 2010*), and PYR, which targets *Pf*DHFR-TS (*Aroonsri et al., 2016*).

In order to assess the feasibility of *Pf*DHS as a target for rationally designed inhibitors, structural study was performed. A three-dimensional structure of *Pf*DHS was constructed by homology modeling using a human DHS crystal structure (Fig. 5). The quality of the *Pf*DHS structure core is estimated to be high with most residues showing QMEAN scores greater than 0.7. The periphery of the *Pf*DHS structure is of lower estimated quality owing to inserts not present in the human DHS protein (Fig. S12). Ligand binding site analysis identified two binding sites with significantly high scores. The top-scoring site with a site score of 1.16 corresponds to the putative substrate binding pocket occupied by spermidine or a competitive inhibitor such as GC7. The tetramer substrate binding pockets are arranged in a sandwich homodimer manner with two sites per homodimer interface (Fig. 6A). The substrate binding pockets in each *Pf*DHS subunit are connected in a tunnel-like fashion highly similar to human DHS (Fig. 6B). Comparison of residues in the vicinity of the substrate binding pocket between *Pf*DHS and human DHS revealed conservation of key interacting residues in *Pf*DHS including Asp368, Asn420, Gly442, Ser443, and Glu451 (Fig. 6C). SiteMap also identified other residues conserved between human and *Pf*DHS that may interact with GC7, or contribute to the environment favorable for GC7 binding (Figs. 6D and 6E). SiteMap detected a second ligand binding site in *Pf*DHS with a site score of 1.01. This site is located in-between the two insertion loops of *Pf*DHS and adjacent to a putative ball-and-chain motif (Fig. S13). The ball-and-chain motif regulates access to the substrate binding pocket, and becomes disordered upon substrate or inhibitor binding (*Liao et al., 1998*; *Umland et al., 2004*).

Hypusinated eIF5A is thought to be important for translation elongation at poorly translated codons, especially consecutive proline-coding codons (*Dever, Gutierrez & Shin, 2014*).

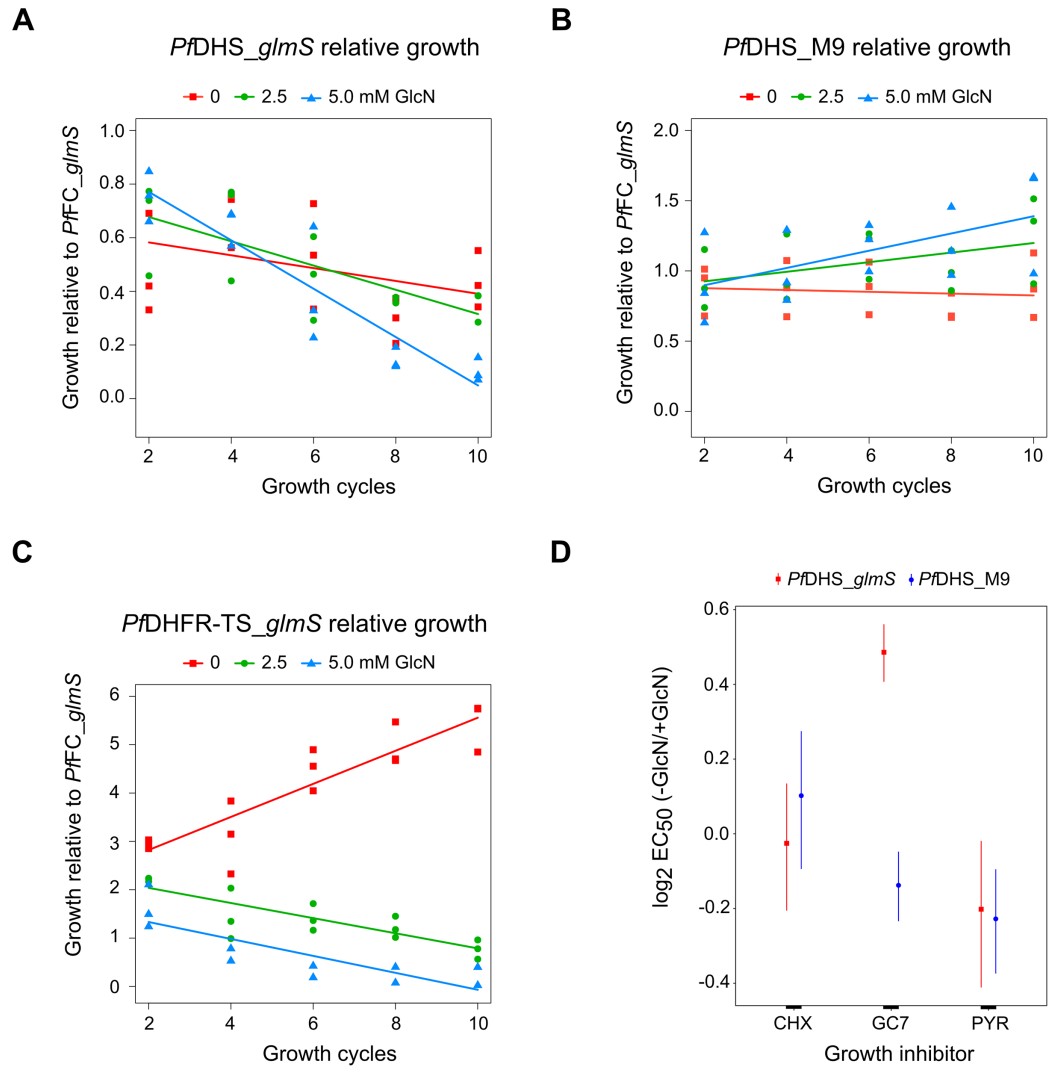

**Figure 4 Growth phenotypes of transgenic parasites.** Effects of glucosamine (GlcN) on the growth of transgenic parasites in co-culture experiments are shown in parts (A–C). Test transgenic parasites *Pf*DHS_*glmS* (A), *Pf*DHS_M9 (B) and *Pf*DHFR-TS_*glmS* (C) were combined with control *Pf*FC_*glmS* parasites in a 1:1 ratio and co-cultured. Samples were taken from the culture every two growth cycles until the tenth cycle. The ratio of test:control transgenic parasite at each time-point was determined by qPCR using primers specific for each transgenic parasite. This ratio was taken as the relative growth value for modelling. The data from three independent experiments for each condition are shown. The lines on the graphs are the linear mixed effect models of growth at the indicated treatment doses of GlcN. *P*-values from likelihood ratio test: 0.0024 (*Pf*DHS_*glmS*); 0.19 (*Pf*DHS_M9); 3.9e-11 (*Pf*DHFR-TS_*glmS*). Part (D) shows the effect of GlcN on modulating transgenic parasite sensitivity to growth inhibitory compounds. Dose-response assays were performed with (2.5 mM) or without GlcN co-treatment. Three or four independent experiments were performed for each condition. The growth inhibitory compounds tested included cycloheximide (CHX), $N^1$-guanyl-1,7-diaminoheptane (GC7) and pyrimethamine (PYR). The 50% inhibitory concentration of each compound ($EC_{50}$) was determined by analysis of dose-response data for *Pf*DHS_*glmS* and *Pf*DHS_M9 transgenic parasites. The dose-response data and fitted curves are shown in Fig. S11. The estimated $\log_2$ ratio of $EC_{50}$ between the −GlcN and +GlcN conditions together with $CI_{95}$ is shown for each compound and transgenic parasite. *P*-values comparing $\log_2 EC_{50}$ (−GlcN/+GlcN) estimates: 0.76 (*Pf*DHS_*glmS*, CHX); 2.0e-15 (*Pf*DHS_*glmS*, GC7); 0.030 (*Pf*DHS_*glmS*, PYR); 0.28 (*Pf*DHS_M9, CHX); 0.0031 (*Pf*DHS_M9, GC7); 9.0e-4 (*Pf*DHS_M9, PYR).

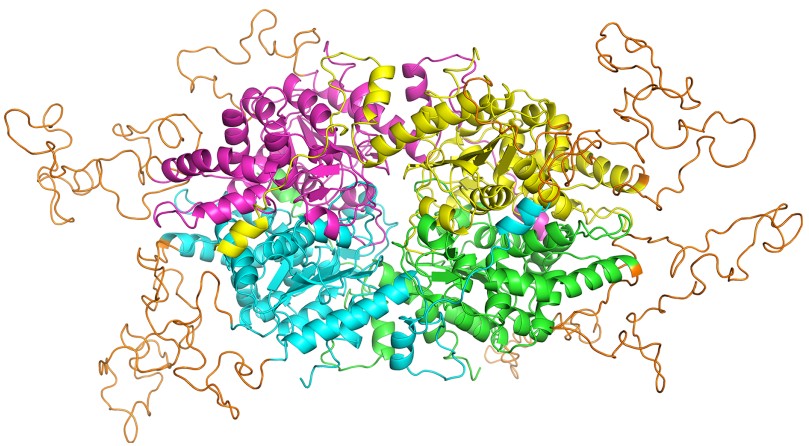

**Figure 5 Homology model of *Pf* DHS structure.** The modeled *Pf* DHS structure is a tetramer of subunits. The subunit cores are colored in magenta, yellow, cyan, and green. *Pf* DHS insertion loops with no homologous residues in human DHS are colored in orange. The alignment of *Pf* DHS with human DHS template (PDB: 1RLZ) is shown in Fig. S12.

Polyproline motifs are numerous in unicellular eukaryotes such as *Saccharomyces cerevisiae* yeast with 769 genes containing PPP or PPG coding motifs (*Mandal, Mandal & Park, 2014*). Moreover, 136/1110 of yeast essential genes (*Zhang, 2004*) contain polyproline coding motifs. The essential genes with polyproline motifs therefore represent 136/5175 (2.6%) of the total yeast protein-coding genes. We therefore performed a bioinformatic survey of the *P. falciparum* proteome for proteins with polyproline motifs that may require hypusinated eIF5A for optimal synthesis following a previous rationale (*Mandal, Mandal & Park, 2014*). 257 proteins with PPP or PPG motifs were found (Table S2), of which insertions of *piggyBac* transposon are not tolerated in the corresponding coding regions for 128 genes (*Zhang et al., 2018*). Using the rationale that *P. falciparum* genes devoid of *piggyBac* transposon insertions are essential, 128/2680 essential genes thus contain polyproline-coding motifs. The essential genes with polyproline motifs in *P. falciparum* therefore represent a similar proportion of the total protein coding genes (128/5305; 2.4%) as in yeast. Among the 142 orthologous genes also with polyproline coding motifs in *P. berghei*, 58 are essential, 16 cause slow growth when disrupted, and 18 are dispensable (*Bushell et al., 2017*).

## DISCUSSION

Hypusine modification of eIF5A is essential in different eukaryotic organisms. In this study, we have validated the DHS enzyme, which is responsible for the first step in the hypusination pathway, as an antimalarial target in *P. falciparum*. We created transgenic parasites with modification of the *Pf* DHS gene for phenotypic study. The modification included a C-terminal GFP tag, which facilitated monitoring of *Pf* DHS protein. Confocal microscopy revealed a cytoplasmic localization of GFP-tagged *Pf* DHS protein, which is consistent with the localization of the mammalian DHS protein (*Sievert et al., 2012*). The cytoplasmic localization may be important to localize eIF5A protein, as DHS-knockout mice show accumulation of nuclear eIF5A (*Pällmann et al., 2015*).
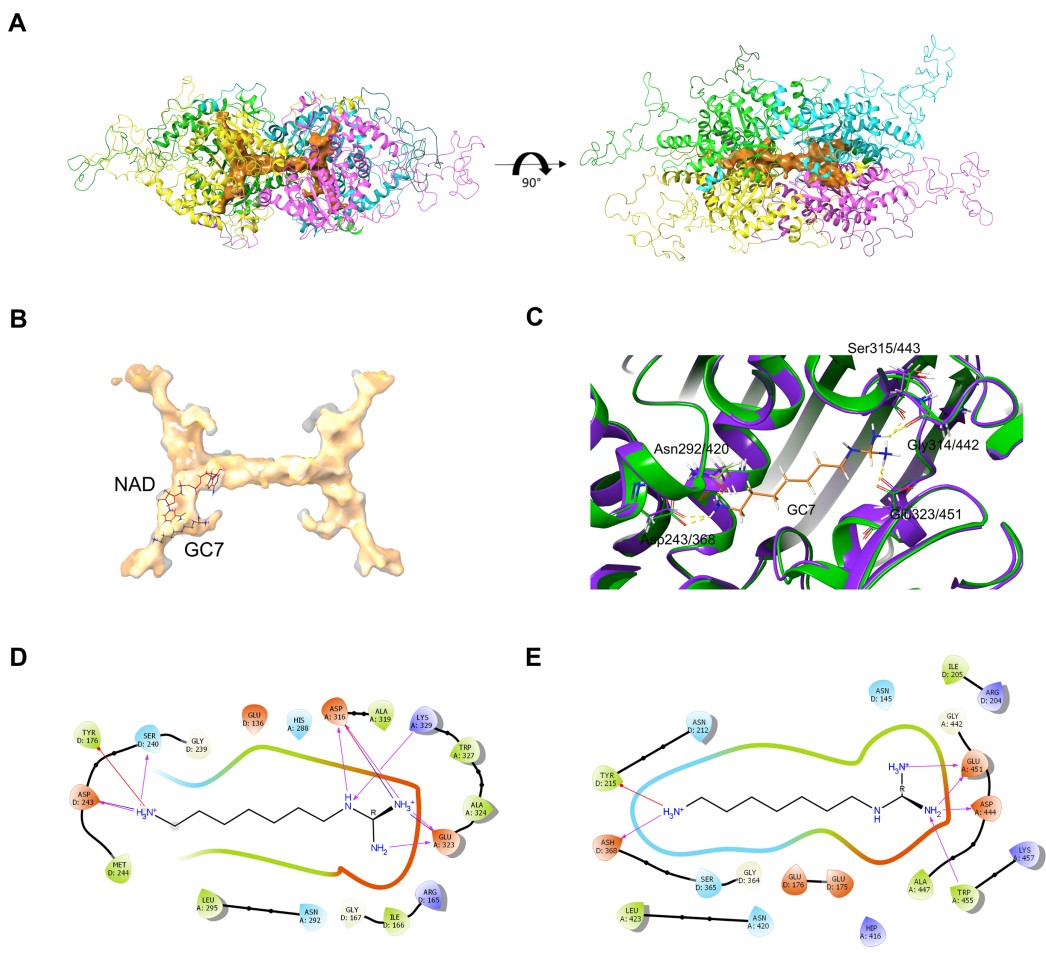

**Figure 6** *Pf* DHS putative substrate binding pocket and GC7 interaction. (A) *Pf* DHS homology model with the top scoring ligand binding site identified by SiteMap in orange. *Pf* DHS tetramer subunits are colored in magenta, yellow, cyan, and green. An alternative view of the same structure rotated 90 degrees is shown on the right. (B) Superposition of the human DHS (PDB: 1RQD) substrate binding pocket (gray) with *Pf* DHS top scoring ligand binding site identified by SiteMap (orange). The NAD and GC7 molecules co-complexed with human DHS are shown for one subunit. (C) Superposition of the human DHS (PDB: 1RQD) GC7 binding domain (green) and corresponding region in *Pf* DHS (purple). Key conserved DHS residues interacting with GC7 are shown, with numbering of human DHS residues followed by the *Pf* DHS counterpart. (D) Interaction map of human DHS and (E) *Pf*DHS residues in the GC7 binding domain. Residues are numbered according to which subunit (A–D) they belong to. Interactions between residues and GC7 are shown by arrows. Red arrow depicts π cation, blue arrows salt-bridge and magenta arrows hydrogen bond interactions.

We were able to attenuate *Pf* DHS expression approximately fivefold in the *Pf* DHS_*glmS* parasite with similar efficiency to that obtained for other essential genes using the same reverse-genetic system (*Prommana et al., 2013*; *Sleebs et al., 2014*; *Elsworth et al., 2014*; *Xie et al., 2016*; *Counihan et al., 2017*; *Thériault & Richard, 2017*). The attenuation of *Pf* DHS expression in this transgenic parasite is specifically induced by GlcN treatment, since GlcN does not reduce expression of the same gene in the *Pf* DHS_M9 parasite, which differs only by two nucleotide substitutions at the *glmS* riboswitch cleavage site that nullify RNA self-cleavage (*Winkler et al., 2004*).

Attenuation of *Pf*DHS expression in the *Pf*DHS_*glmS* causes a growth defect, although the decline of *Pf*DHS_*glmS* parasites in culture is less rapid than *Pf*DHFR-TS-attenuated parasites. These data suggest that *Pf*DHS is essential and explain why no insertions of *piggyBac* transposon are tolerated in this gene (*Zhang et al., 2018*). However, inducible null mutants (e.g., created by using the DiCre inducible knockout method (*Collins et al., 2013*)) are required for definitive proof of essentiality. The slow decline of *Pf*DHS-attenuated parasites suggests that the residual hypusinated protein is sufficient to support viability for at least one growth cycle, perhaps because hypusinated protein is long-lived. In support of this explanation, the level of hypusinated protein in *Pf*DHS-attenuated parasites is modestly reduced. A total of 60% reduction of hypusinated eIF5A in a conditional mutant of yeast with attenuated DHS expression is deleterious (*Galvão et al., 2013*), suggesting that a certain level of hypusinated eIF5A is necessary for eukaryote cell growth. Alternatively, the growth defect in *Pf*DHS-attenuated parasites could be due to loss of hypusination and/or a "moonlighting" function of *Pf*DHS. To test possible "moonlighting" functions of *Pf*DHS, data from catalytically inactive *Pf*DHS mutants are required. Although the slow decline of *Pf*DHS-attenuated parasites could be attributed to incomplete attenuation of *Pf*DHS expression, decline of null *Pf*DHS mutants in growth assays may not be much more rapid since null DHS mutants of yeast can undergo several cell divisions before arrest (*Park, Joe & Kang, 1998*).

The competitive growth assay developed in this study is suitable for monitoring of growth over extended periods and demonstrating latent (more than two growth cycles) growth defects. Our method has the advantage that no correction for dilution factors is necessary, which could introduce substantial error. However, we have not performed head to head comparisons of the competitive growth assay with other methods to assess accuracy. The competitive growth assay could be scaled-up by pooling parasites with *glmS* riboswitch modifications of different genes and monitoring the growth of each mutant by next-generation sequencing, similar to that described for mutants carrying *piggyBac* insertions (*Bronner et al., 2016*). The competitive growth assay is an alternative to the plaque-based growth assay, which has been used to demonstrate latent growth defects in essential invasion genes (*Thomas et al., 2016*). The small size of plaques make quantification difficult, especially if a scanner of sufficiently high resolution is not available. Slow growing mutant parasites may also have failed to reach a sufficiently critical mass at the end of the assay period such that no plaque is visible and the growth defect over-estimated. The growth defects of mutant parasites inferred from plaque assay can be confounded by competition with wild-type parasites (*Lehmann et al., 2018*). Furthermore, the plaque-based growth assay is an end-point assay and so cannot provide information of growth dynamics like the competitive growth assay.

From the latent growth defect in *Pf*DHS mutants, *Pf*DHS can be considered a less vulnerable antimalarial target than other essential genes such as *Pf*DHFR-TS. Information of target vulnerability is important for antimalarial development. It may be difficult to develop potent antimalarial derivatives from hit compounds obtained by high throughput screening if the target has a low vulnerability. This is because inhibitors of less vulnerable targets need to bind the target with very high affinity to ensure that almost

all target activity is inhibited at therapeutic doses for antimalarial efficacy. In contrast, compounds against the most vulnerable targets (for which even small reductions in activity are deleterious to the parasite) may only need to have moderate binding affinity for high antimalarial efficacy. Furthermore, knowledge of target vulnerability could be important for consideration of how easily resistance could evolve. Resistance mutations in less vulnerable targets that cause a small reduction of inhibitor binding affinity could render drugs ineffective. If a panel of mutants with the expression of different essential genes attenuated using the *glmS* riboswitch or another reverse genetic tool was available, competitive growth assays could be performed to systematically compare target vulnerabilities and triage targets for antimalarial development.

*Pf*DHS-attenuated parasites are significantly more sensitive to growth inhibition by GC7, suggesting that the primary in vivo target of this compound is *Pf*DHS. The slow decline of genetically attenuated *Pf*DHS parasites in competitive growth assays suggests that some of the antimalarial effect of GC7 observed in standard antimalarial assays, in which growth is assessed over shorter periods, could be attributed to inhibition of secondary targets (off-targeting). The $\log_2$ $EC_{50}$ (–GlcN/+GlcN) ratio for GC7 against the *Pf*DHS_*glmS* parasite is markedly lower than the ratios for antifolates against the *Pf*DHFR-TS_*glmS* parasite (*Aroonsri et al., 2016*). Low $EC_{50}$ ratios are consistent with off-targeting. In support of this inference, GC7 has reported off-target effects in human cells, including induction of autophagy independently of eIF5A activity (*Oliverio et al., 2014*). Alternatively, the low $EC_{50}$ ratio for GC7 against the *Pf*DHS_*glmS* parasite may not be due to off-targeting if the level of *Pf*DHS activity remaining in the +GlcN condition is still in excess of what is required for growth.

The inference that *Pf*DHS is the primary antimalarial target of GC7 is supported by the in silico modeling data, which show conservation of substrate binding pocket and *Pf*DHS residues putatively involved with GC7 interaction. The high conservation of substrate binding pocket could make the design of antimalarials specific to the *Pf*DHS target challenging. However, the modeled *Pf*DHS structure reveals a second potential ligand binding site that could also be exploited for rational drug design. The second site overlaps insert residues not present in human DHS and a putative ball-and-chain motif. Compounds binding to this site thus could act as allosteric inhibitors by preventing access of protein substrates to the substrate binding pocket for deoxyhypusine modification. *X*-ray structural data of *Pf*DHS could provide important insights into *Pf*DHS for drug design not revealed by in silico modeling, including the roles of the ball-and-chain motif and inserts peripheral to the core.

The primary function of hypusinated eIF5A for translation elongation, particularly of polyproline motifs may be conserved in *P. falciparum*, as a number of its proteins contain these motifs. Although *P. falciparum* has markedly fewer polyproline-motif proteins than yeast, the number of these proteins that are essential with respect to the total protein complement is similar. Insufficient synthesis of one or more essential protein could be responsible for the growth defect in *Pf*DHS attenuated mutants, although proteomic data are needed to test this hypothesis. The small number of polyproline motif proteins

in *P. falciparum* that potentially require hypusinated *Pfe*IF5A for their production could explain why global protein synthesis (in the short term) is not significantly reduced when *Pf*DHS function is attenuated.

## CONCLUSIONS

The loss of *Pf*DHS function leads to a growth defect, suggesting that this gene is essential. However, definitive proof from null mutants is required to conclude essentiality. Loss of *Pf*DHS function leads to reduction of hypusination, which may affect synthesis of some proteins. *Pf*DHS is not as vulnerable a target as other essential genes such as *Pf*DHFR-TS, although it can be targeted by antimalarials such as GC7.

### Funding

We received funding from the Thailand Research Fund grant nos. RSA5780007, RSA5880064, and TRG6080001 to Philip. J. Shaw, Chairat Uthaipibull and Aiyada Aroonsri respectively; National Science and Technology Development Agency (NSTDA) (Thailand) project nos. P1450752, P1300832, P1450883, P1751076 to Philip. J. Shaw, Chairat Uthaipibull, Sumalee Kamchonwongpaisan and Aiyada Aroonsri, respectively; NSTDA Core Researcher Grant no. P1850116 to Sumalee Kamchonwongpaisan and BIOTEC (Thailand) Young Fellow Research Grant no. P1750543 to Aiyada Aroonsri. The funders had no role in study design, data collection and analysis, decision to publish, or preparation of the manuscript.

### Grant Disclosures

The following grant information was disclosed by the authors:
Thailand Research Fund: RSA5780007, RSA5880064, and TRG6080001.
National Science and Technology Development Agency (NSTDA) (Thailand) project:
P1450752, P1300832, P1450883, P1751076.
NSTDA Core Researcher: P1850116.
BIOTEC Young Fellow Research Grant: P1750543.

### Competing Interests

The authors declare that they have no competing interests.

### Author Contributions

- Aiyada Aroonsri conceived and designed the experiments, performed the experiments, analyzed the data, prepared figures and/or tables, authored or reviewed drafts of the paper, approved the final draft.
- Navaporn Posayapisit conceived and designed the experiments, performed the experiments, analyzed the data, prepared figures and/or tables, authored or reviewed drafts of the paper, approved the final draft.
- Jindaporn Kongsee performed the experiments, approved the final draft.
- Onsiri Siripan performed the experiments, approved the final draft.

- Danoo Vitsupakorn conceived and designed the experiments, performed the experiments, prepared figures and/or tables, approved the final draft.
- Sugunya Utaida performed the experiments, authored or reviewed drafts of the paper, approved the final draft.
- Chairat Uthaipibull analyzed the data, authored or reviewed drafts of the paper, approved the final draft.
- Sumalee Kamchonwongpaisan analyzed the data, authored or reviewed drafts of the paper, approved the final draft.
- Philip J. Shaw conceived and designed the experiments, performed the experiments, analyzed the data, prepared figures and/or tables, authored or reviewed drafts of the paper, approved the final draft.

### Human Ethics

The following information was supplied relating to ethical approvals (i.e., approving body and any reference numbers):

Human erythrocytes were obtained from donors after providing informed written consent, following a protocol approved by the Ethics Committee, National Science and Technology Development Agency, Pathum Thani, Thailand, document no. 0021/2560.

### Data Availability

The raw data are included in the figures and Supplemental Files.

### Supplemental Information

Supplemental information for this article can be found online at http://dx.doi.org/10.7717/peerj.6713#supplemental-information.

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
