# Peer review of "Validation of Plasmodium falciparum deoxyhypusine synthase as an antimalarial target"

_PeerJ, doi:10.7717/peerj.6713_

## Round 0.1 · original submission · Major Revisions

The review process is now complete, and three thorough reviews from highly qualified referees are included at the bottom of this letter. All reviewers including myself agree the manuscript is well written and deserves to be published. Although there is considerable merit in your paper, we also identified some concerns that must be considered in your resubmission. Please, provide a balanced discussion of the study’s results avoiding overstated conclusions.

Reviewer 1 ·

Basic reporting

In this manuscript, the investigators evaluate whether hypusination, a post-translational protein modification, is essential in P. falciparum malaria parasites. Using a glmS riboswitch, the investigators reduce expression of PfDHS, an enzyme required for hypusination. Immunoblot analysis indicates a significant reduction in PfDHS-GFP levels, although protein is still present. Hypusination levels are modestly but reproducibly decreased to 30% of untreated levels (Fig. 3), even with maximum reduction of PfDHS-GFP protein levels. With maximum reduction of PfDHS-GFP protein level, parasite morphology remains grossly normal. Using a competitive growth assay and comparing relative parasite levels by qPCR, the investigators conclude that reduced DHS protein levels cause a modest, albeit significant, reduction in asexual parasite replication. Overall, this is a clearly written manuscript that provides initial evidence on the biological function of a possibly essential gene in malaria parasites. In general, replicates are well described and adequate, and analyses are transparent and statistically rigorous. There are a number of minor concerns noted. Specifically, several conclusions are more strongly stated than is warranted by the data presented. In addition, because the growth defect is so modest and the data presented has larger-than-expected biological variability, additional data should be provided in support, as noted below. Specific comments are noted below:

Experimental design

1) Fig 4a. I understand why the investigators chose to use growth competition given the modest effect on parasite growth by reduced DHS. However, the data presented in Fig 4A is not quite convincing. The biological variability of the 5 mM GlcN samples (see t=6 in particular) is wide, and substantially larger variability is seen for this strain than, for example, DHFR (Fig 4C). I would prefer to see the data in terms of % parasitemia. The investigators suggest that this method is preferred over correcting for dilution factors, but it would be preferable to present those data side-by-side for direct comparison.

2) In addition, I suggest softening the language, as the data presented is promising but not sufficiently definitive to conclude that “PfDHS gene is essential” (line 548). Certainly, PfDHS is not a highly promising drug target, if >5-fold reduction in activity leads to only a modest growth defect after multiple cycles of asexual growth.

3) The investigators should also soften the language linking the phenotypic effects of reduced DHS levels to reduced hypusination. Many metabolic enzymes have “moonlighting” functions that do not depend on their catalytic activity, and therefore the cellular function of DHS may be broader than just hypusination. A better control would be integration of a “catalytically dead” version of this protein, so that enzymatic activity and hypusination could be distinguished from any other cellular roles.

4) The investigators should soften the language with respect to GC7. Specifically, the data presented does not definitively confirm the in vivo target of GC7 (line 522-523). Although this compound appears to inhibit DHS in vitro, it is not clear that its antiparasitic effects are mediated exclusively through DHS. Since “off target” effects are possible (even likely at high concentrations), sensitivity to this compound is suggestive but far from definitive.

Validity of the findings

As noted above, conclusions are somewhat overstated and alternative explanations for findings have not been discussed.

Reviewer 2 ·

Basic reporting

The paper is clearly written for a general audience, provides adequate context and motivation for the work done, and includes legible and informative figures/tables to support the manuscript text.

Experimental design

The central research questions related to whether dehydrohypusine in Plasmodium plays an essential role and its potential as an antimalarial drug target are clearly stated. Appropriate experiments explained in sufficient detail to allow repeating are carried out with adequate rigor. Prior work on this gene in Plasmodium has been published, but direct information on its likely essentiality had not been reported until this manuscript.

Validity of the findings

Experimental data is accompanied by appropriate controls--e.g. use the functional glmS ribozyme for controlling gene expression is paired with a control using an inactive version of this ribozyme, and these are used together throughout the manuscript.

Additional comments

Overall, the manuscript is clearly written and provides useful additional findings to the existing literature.

Two minor points to address are:

1) In the Results and again the Discussion sections (e.g lines 446-452 ), the authors attempt to justify why the PfDHS enzyme is more likely essential based on the frequency of proteins with poly-Pro motifs in various proteins (including non-essential genes. This could be true. At the same time, though, if only one essential poly-Pro-containing protein were dependent on DHS function this enzyme would also be essential. Thus, a more balanced (though not necessarily exhaustive) discussion exploring why PfDHS function could be essential should be considered.

2) Lines 527-528. The statement “Low EC50 rations is consistent with off-targeting” is not always true and should be revised for improved accuracy. An alternative explanation, for example, could be that target protein expression levels after knockdown relative to what is needed for parasite survival is modest, and similar compound concentrations are required to abrogate function of the target protein

Reviewer 3 ·

Basic reporting

Line 381: It is not clear what is the puromycilation signal. I suggest to add one sentence for explanation.

Line 532-535: The sentence "this number is markedly fewer than in yeast (Li et al., 2014) and other lower eukaryotes (Mandal, Mandal & Park, 2014)." is ambiguous. Provide the number of essential proteins containing polyproline motifs per total protein numbers or its percentage in yeast and other lower eukaryotes clearly so that readers are able to compare. Also are they significantly different between Plasmodium and others?

To discuss if the parasite molecule is a potential target of drug development, structural difference between parasite and human molecules would provide important insights. This pathway is well-conserved among eukaryotes, so discuss the feasibility of DHS as a drug target based on the structural dissimilarity or similarity in the discussion section.

I strongly recommend authors to supply legends for supplementary figures. Currently nothing is provided. Some are impossible to understand without the legend.

Experimental design

Line 381 and Fig S1C: Fig. S1C is not really sufficient to show the expected integration of the plasmid into the target gene locus. Southern blot analysis is required to show the integration to the DHS gene locus (glmS and M9) as was done for PfFC gene locus in supplementary Fig. S2. I guess the pattern of the lane 3 on the Southern blot in Fig S2 indicates disrupted gene locus and existence of the remaining episomal plasmid. Remaining plasmid could produce false positive by PCR.

Line 408. What is the effect of GlcN against the FC expression in PfFC_glmS parasite line? Why Western blot was not performed? Is it possible that the gene locus was disrupted and FC was not expressed anymore? If that is the case, 3D7 strain would serve as a control and I cannot see the rationality to use PfFC_glmS as a control. Comments?

In the conclusions, authors clearly state "Loss of PfDHS function leads to reduction of eIF5A hypusination,..". However, the claim that < 20-kDa band stained with anti-hypusine antibody was eIF5A was solely speculated based on the band size. Immunoprecipitation with anti-hypusine antibody and staining with anti-eIF5A antibody or mass-spec analysis with proper controls is suggested. Otherwise lower the tone.

Validity of the findings

Having a look at Fig. S6, I have a concern that authors might not clearly define the parasite stages; ring, trophozoite, and schizont. For example, "trophozoite"s indicated in A and C are actually schizonts as they have multiple nuclei. Also "ring" indicated in B is a trophozoite as this clearly contains hemozoin. This misclassification likely reflect analysis based on the stages, I suggest authors re-examine all images, re-classify the parasites for re-analysis.

Additional comments

Line 538-544 I feel there is a gap in the following argument: "P. falciparum also possesses a number of non-essential polyproline-motif proteins, including notably the erythrocyte membrane protein 1 family, which plays important roles in virulence (Deitsch & Hviid, 2004). Interestingly, polyproline motifs are also abundant in the variable surface glycoprotein (VSG) protein family of the African sleeping sickness parasite Trypanosoma brucei. Translation of VSG proteins is sensitive to eIF5A levels (Nguyen et al., 2015), possibly pointing to a similar mechanism for controlling expression of virulence factors among these highly diverged human parasites.". Polyproline residues may have structural function such as forming polyproline helices, which may be important for the protein-protein interaction, as a linker between domains in newly generated species-specific proteins, and so on. Any evidences suggest T. brucei use hypusinated eIF5 to control VSG expression? Do authors suggest involvement of eIF5 of T. brucei and P. falciparum in the antigenic variation? Unless authors provide more robust data or more comprehensive bioinformatics analysis including other eukaryotic pathogens, I would suggest to remove these sentences.

---

## Round 0.2 · Minor Revisions

Please, address the point raised by the Reviewer #3.

Reviewer 1 ·

Basic reporting

No comment

Experimental design

No comment

Validity of the findings

No comment

Additional comments

The authors have addressed my previous concerns with this manuscript and I have no new concerns.

Reviewer 3 ·

Basic reporting

no comment

Experimental design

The manuscript is significantly improved. However, I have still one concern regarding Fig. S2. I agree with the authors about the identity of the 6-kb band in Fig. S2C lane 3. If so, it is not recommended to show in Fig. S2B the expected modified gene locus by design, but not by the obtained data. Author should draw a schematic diagram including concatenated plasmid part for this transgenic parasite (an example is attached). The identity of this 6-kb band should be explained in the figure legend, too.

Validity of the findings

no comment

Additional comments

no comment

Annotated reviews are not available for download in order to protect the identity of reviewers who chose to remain anonymous.

---

## Round 0.3 · accepted · Accept

The authors have made all changes required.

#